# Exploring the Potential of Halotolerant Actinomycetes from Rann of Kutch, India: A Study on the Synthesis, Characterization, and Biomedical Applications of Silver Nanoparticles

**DOI:** 10.3390/ph17060743

**Published:** 2024-06-06

**Authors:** Paras Dayma, Nisha Choudhary, Daoud Ali, Saud Alarifi, Pravin Dudhagara, Kuldeep Luhana, Virendra Kumar Yadav, Ashish Patel, Rajesh Patel

**Affiliations:** 1Department of Biosciences, Veer Narmad South Gujarat University, Surat 395007, Gujarat, India; paras_dayma@yahoo.com (P.D.); dudhagarapr@gmail.com (P.D.); 2Department of Life Sciences, Hemchandracharya North Gujarat University, Patan 384265, Gujarat, India; nishanaseer03@gmail.com; 3Department of Zoology, College of Science, King Saud University, P.O. Box 2455, Riyadh 11451, Saudi Arabia; 4Department of Biotechnology, Hemchandracharya North Gujarat University, Patan 384265, Gujarat, India; kuldeep.dnatech@gmail.com

**Keywords:** actinomycete, halotolerant, silver nanoparticles, multidrug resistance, Rann of Kutch

## Abstract

A tremendous increase in the green synthesis of metallic nanoparticles has been noticed in the last decades, which is due to their unique properties at the nano dimension. The present research work deals with synthesis mediated by the actinomycete *Streptomyces tendae* of silver nanoparticles (AgNPs), isolated from Little and Greater Rann of Kutch, India. The confirmation of the formation of AgNPs by the actinomycetes was carried out by using a UV-Vis spectrophotometer where an absorbance peak was obtained at 420 nm. The X-ray diffraction pattern demonstrated five characteristic diffraction peaks indexed at the lattice plane (111), (200), (231), (222), and (220). Fourier transform infrared showed typical bands at 531 to 1635, 2111, and 3328 cm^−1^. Scanning electron microscopy shows that the spherical-shaped AgNPs particles have diameters in the range of 40 to 90 nm. The particle size analysis displayed the mean particle size of AgNPs in aqueous medium, which was about 55 nm (±27 nm), bearing a negative charge on their surfaces. The potential of the *S. tendae*-mediated synthesized AgNPs was evaluated for their antimicrobial, anti-methicillin-resistant *Staphylococcus aureus* (MRSA), anti-biofilm, and anti-oxidant activity. The maximum inhibitory effect was observed against *Pseudomonas aeruginosa* at (8 µg/mL), followed by *Escherichia coli* and *Aspergillus niger* at (32 µg/mL), and against *Candida albicans* (64 µg/mL), whereas *Bacillus subtilis* (128 µg/mL) and *Staphylococcus aureus* (256 µg/mL) were much less sensitive to AgNPs. The biosynthesized AgNPs displayed activity against MRSA, and the free radical scavenging activity was observed with an increase in the dosage of AgNPs from 25 to 200 µg/mL. AgNPs in combination with ampicillin displayed inhibition of the development of biofilm in *Pseudomonas aeruginosa* and *Streptococcus pneumoniae* at 98% and 83%, respectively. AgNPs were also successfully coated on the surface of cotton to prepare antimicrobial surgical cotton, which demonstrated inhibitory action against *Bacillus subtilis* (15 mm) and *Escherichia coli* (12 mm). The present research integrates microbiology, nanotechnology, and biomedical science to formulate environmentally friendly antimicrobial materials using halotolerant actinomycetes, evolving green nanotechnology in the biomedical field. Moreover, this study broadens the understanding of halotolerant actinomycetes and their potential and opens possibilities for formulating new antimicrobial products and therapies.

## 1. Introduction

Actinomycetes are a diverse category of Gram-positive, filamentous bacteria that have higher guanine + cytosine content [1]. It is widely distributed in terrestrial and aquatic ecosystems and plays a significant biological and ecological role in the environment, including the degradation and recycling of substances, and the generation of bioactive molecules [2,3]. Prokaryotes and eukaryotes both can produce antimicrobial compounds whereas the prokaryotic group of organisms, actinomycetes are the prolific producers of bioactive substances [4]. Approximately 70% of the total antibiotics that are used today for the treatment of various diseases are alone obtained from actinomycetes, especially *Streptomyces* [5] in addition to taromycin A [6], retimycin A [7,8], and tetrocarcins [9,10]. *Streptomyces*, a well-known genus, which produces diverse bioactive secondary metabolites like antibiotics, anticancer [11], antiviral, anti-malarial agents [12,13], etc. contributing significantly to medically important antibiotics [14,15], while other unexplored actinomycetes genera hold potential for bioactive metabolites [16,17]. In the last decade, microbial infections have increased tremendously [18,19,20], leading to antibiotic resistance due to misuse [21,22,23]; resistance mechanisms include genetic changes and metabolic pathways [24], demanding the formulation of new compounds to combat multidrug resistance [25,26,27].

The high salinity and alkalinity are the uniqueness of the desert, which leads to distinct microbial compositions and structures [28]. The defense potential of these microorganisms is exhibited by producing bioactive molecules against drug-resistant bacteria [29,30]. These microorganisms have the potential for antibiotic production and nanoparticle synthesis, increasing stability and biocidal action through surface modification by biological macromolecules [31,32].

Noble metallic nanoparticles (NPs) like silver, gold, and platinum, synthesized biologically, are biocompatible [33,34]. These NPs are effective against drug-resistant bacteria, which offers safer solutions for biomedical applications [35]. Biocompatibility remains a major issue among the NPs synthesized by chemical and physical routes. Recently silver nanoparticles (AgNPs) have been widely used in biomedicine, textiles, etc. due to their effectiveness and antimicrobial activity [36]. To date, several types of microorganisms have been used for the synthesis of AgNPs like bacteria [37], fungi [38], algae [39], and actinomycetes. However halotolerant actinomycetes along with antimicrobial activity have been rarely used for the synthesis of AgNPs, Nevertheless, actinomycetes have received relatively a little attention regarding their potential for synthesizing nanoparticles [40]. Synthesis and characterization of AgNPs [41] have been documented from *Streptomyces* sp., *Nocardia sp.* [42,43,44], *Thermomonospora sp.* [45], and *Rhodococcus sp.* Among *Streptomyces, Streptomyces sp.* VITBT7, *Streptomyces. sp.* MBRC-91, *Streptomyces parvulus* SSNP11, *Streptomyces rochei, Streptomyces aureofaciens* MTCC356, *Streptomyces spp.* I, II; *Streptomyces albidoflavus* CNP10, *Streptomyces parvulus* DPUA1549, *Streptomyces seoulensis* DPUA 1747, *Streptomyces owasiensis* DPUA 1748, *Streptomyces spp.* 09 PBT 005, *Streptomyces narbonensis* SSHH-1E and *Streptomyces xinghaiensis* OF1 [46] have been explored to date for the formation of AgNPs.

Wypij et al. synthesized AgNPs by using an acidophilic actinobacterial SH11 strain isolated from pine forest soil which was polydisperse and spherical in shape whose mean size was 13.2 nm. The AgNPs exhibited antibacterial activity in *Staphylococcus aureus*, *Bacillus subtilis*, and *Escherichia coli* by using the disc diffusion minimum inhibitory concentration (MIC) method [47]. Otari et al. also synthesized 5–50 nm AgNPs by using *Rhodococcus* spp. and assessed their antimicrobial activity against pathogenic microorganisms [48]. Recently, Enain synthesized AgNPs from *Streptomyces avermitilis* Azhar A.4 E and assessed their larvicidal effect on the black cutworm, *Agrotis ipsilon* [49].

Out of all these investigations, hardly any attempts were made at the synthesis of AgNPs from the halotolerant actinomycetes isolated from the desert area. So, the current investigation reports the synthesis of AgNPs from desert-isolated halotolerant actinomycetes which also have activity against methicillin-resistant *Staphylococcus aureus* (MRSA) pathogens. 

The major aim of this current study was to synthesize AgNPs using actinomycetes isolated from the salty desert of Kutch, i.e., Rann of Kutch. One of the objectives was to isolate and characterize the halotolerant actinomycetes from the saline desert area. The second objective was to observe the antibiotic production and antimicrobial activity profile of the isolated strain. Thirdly, we aimed to screen, identify, and design applications for the development of AgNPs from the potential strain of actinomycetes. Another objective was to characterize the developed AgNPs by using analytical instruments. Lastly, our goal was to evaluate the biomedical applications, including the antimicrobial activity of the AgNPs. Such a greener approach will play a crucial role in material science and it will also prove valuable in combating drug-resistant bacteria.

## 2. Results 

### 2.1. Extracellular Biosynthesis of AgNPs from Saline Desert Actinomycetes

For the reduction of Ag^+^ to Ag^0^, different ratios of cell-free supernatant to AgNO_3_ were examined. The reaction mixture, having an equal amount of AgNO_3_ and culture supernatant, showed a rapid color change compared to other reactions. The synthesis of AgNPs using actinomycete was examined mainly through the observation of the color change in the reaction mixture in the presence of 1 mM AgNO_3_. The difference in color change indicates the difference in the size of the AgNPs. Control experiments without actinomycete showed no color change. The transformation of color from pale yellow to dark brown occurred after 4 to 24 h of incubation.

### 2.2. UV-Vis Spectroscopic Analysis of AgNPs Synthesized by S. tendae (GR-CHA-4)

UV-Vis spectra of AgNPs synthesized by actinobacteria and control are shown in Figure 1; these were recorded immediately after the color change (within 1 h) from pale yellow to brown. An uninoculated MGYP medium was used as a control. Homogenous AgNPs have been known to produce a Surface Plasmon Resonance (SPR) band in the range of 420–450 nm [50]. The absorbance peak reflects the size and shape of silver NPs and a shift in the SPR peak to longer wavelengths with an increase in particle size could suggest that AgNPs produced by actinomycete isolate are smaller in size with a spherical shape [51]. The AgNPs synthesized within 4 h represented rapid synthesis [52]. The AgNPs synthesized by *S. tendae* showed an absorption peak at 405 nm, confirming the formation of AgNPs.

### 2.3. Effect of Reaction Parameters on AgNP Synthesis Using Isolate S. tendae (GR-CHA-4)

#### 2.3.1. Medium Optimization for Nanoparticle Synthesis

Based on the literature survey, the nanoparticle-producing ability of the isolate *S. tendae*-GR-CHA-4 was further optimized by using different growth media such as MGYP, ISP Medium No 1, starch casein broth, casein glycerol broth, and M9 minimal medium (sodium acetate as carbon source). The rapid AgNP synthesis was observed after 4 **h** using MGYP. Furthermore, AgNPs did not show clumping out of five media. MGYP was best for AgNP synthesis using isolate GR-CHA-4. The other four media were not suitable for AgNP formation using GR-CHA-4 isolate due to slow synthesis (after 24 h) and the presence of aggregation.

#### 2.3.2. Synthesis of AgNPs under Normal, Alkaline, Saline, and Saline + Alkaline Conditions

To optimize the effect of pH and NaCl concentration together with the nanoparticle production ability of the selected isolate, the isolate was grown on an MGYP medium in the presence of four different conditions, i.e., normal (0.0% *w/v* salt, 7.0 pH), alkaline (0.0% *w/v* salt, 9.0 pH), saline (5.0% *w/v* salt, 7.0 pH), and saline + alkaline (5% *w/v* salt, 9.0 pH) condition. Among all the four conditions used, the combination of 0% *w/v* salt and pH 7 supported the nanoparticle synthesis, while no synthesis occurred in other conditions.

#### 2.3.3. Effect of AgNO_3_ Concentration on Nanoparticle Synthesis 

AgNP formation is highly influenced by the concentration of the parental compound. Low concentration will not produce applicable AgNPs, while high concentration will cause clumping and aggregation. Therefore, optimization is required for the formation of stable AgNPs. The formation of AgNPs took place in the presence of all the AgNO_3_ concentrations starting from 1 mM to 5 mM. From the tested concentrations, 2 mM was found to be the optimum one, as it showed the absorption peak with the highest intensity (Figure 2a).

#### 2.3.4. Effect of pH on AgNP Synthesis 

Variations in the pH of the reaction mixture influence the functionality of the enzymes and metabolites, thereby affecting the synthesis of NPs. This can lead to changes in the morphology and distribution of NPs. To check the relationship between pH and AgNP synthesis, the reaction mixture was allowed to react at pH 5.0, 6.0, 7.0, 8.0, and 9.0. The maximum synthesis of NPs occurred at pH 7.0, followed by pH 8.0 and pH 6.0, which showed much less bioreduction of AgNO_3_ (Figure 2b), while pH 5.0 and pH 9.0 did not support the reduction of AgNO_3_ to AgNPs.

#### 2.3.5. Effect of Temperature on AgNP Synthesis

Biosynthesis of AgNPs was observed at 30 °C, 40 °C, 50 °C, 60 °C, 70 °C, and 80 °C. AgNPs were formulated at all six temperatures, but the rate of synthesis and the yield obtained were different. There was a gradual increase in the velocity of the reaction as well as the concentration of AgNPs from 30 °C to 60 °C, which can be observed from UV-Vis spectra. Above 60 °C, there was a sharp decrease in the synthesis of AgNPs (Figure 2c).

#### 2.3.6. AgNP Synthesis under Optimized Conditions Using *S. tendae* (GR-CHA-4)

All the optimized factors were put together in the experiment to study the nanoparticle synthesis under optimized conditions. The MGYP was used as a growth medium for the growth of the selected strain (GR-CHA-4). The parameters applied during nanoparticle synthesis were pH 7.0, temperature 60 °C, and 2 mM AgNO_3_. After incubation, color change from pale yellow to brown was observed within 1 h. So, based on various factors, the optimized medium is favorable for synthesizing AgNPs on a large scale.

#### 2.3.7. Particle Size Analysis (PSA) and Zeta Potential of Synthesized AgNPs 

The size distribution, average particle size, and concentration of synthesized AgNPs were measured by Nanoparticle Tracking Analysis (Figure 3a). The NTA is based on two properties of particles, light scattering and Brownian motion, and in comparison, to the Dynamic Light Scattering (DLS) method, NTA is more accurate. The results of PSA displayed the mean particle size of about 55 nm (±27 nm). The concentration of AgNPs was 2.70 × 10^8^ particles/mL. The zeta potential (ZP) of the AgNPs was −20.4 mV, while conductivity was 0.0443 mS/cm (Figure 3b).

#### 2.3.8. Fourier Transform Infrared (FT-IR) Spectroscopy of AgNPs for the Identification of Functional Groups 

FTIR spectroscopy was used to identify the surface chemistry of biosynthesized AgNPs. FTIR measurements are also used to study the possible interaction between AgNPs and protein molecules, which is responsible for the synthesis and stabilization of well-dispersed AgNPs in the reaction mixture [53]. Figure 4 shows the presence of four major functional groups recorded between wave numbers 531 and 3328 cm^−1^. 

#### 2.3.9. X-ray Diffraction (XRD) Analysis for Phase Identification 

The exact nature of the AgNPs can be deduced from the XRD analysis. The XRD pattern (Figure 5) revealed the crystalline nature of synthesized AgNPs. The XRD pattern for AgNPs showed diffraction peaks at 27.06, 35.7, 37.8, 40.9, 43.8, 54.1, 56.5, and 68.8°. Out of all these peaks, the highest and sharpest peaks were at 27.06 and 54.1°.

#### 2.3.10. Morphological Analysis of AgNPs Synthesized by *S. tendae* by Using Scanning Electron Microscope (SEM)

The SEM image of the dried AgNPs shows the spherical-shaped NPs in scattered form (Figure 6a). The SEM micrograph clearly showed the presence of spherical-shaped particles having diameters in the range of 40 to 90 nm.

The energy dispersive X-ray spectroscopy (EDS) spectra of AgNPs shown in Figure 6b reveal peaks for Ag, Cl, S, Mg, and Ca. Out of all these elements, Ag was 91.64 (wt.%), Ca was (5.05%), Mg (1.25%), Cl (0.96%), and S (1.1%).

#### 2.3.11. Three-Dimensional Analysis of AgNPs Using an Atomic Force Microscope (AFM) 

AFM allows 2D and 3D characterization of nanomaterials, and the topographical features were analyzed using the AFM microscope for better visualization of surface morphology. AFM analysis provides information about the morphology and surface roughness of the materials. The AFM characterization shows two- and three-dimensional AFM images of AgNPs (Figure 7a). The 2D surface topographical images (Figure 7b) reveal the spherical nature of particles, whereas the 3D AFM micrograph shows the height and surface roughness of the AgNPs.

#### 2.3.12. Antimicrobial Activity of AgNPs (MIC)

Table 1 shows the MIC and MBC values of AgNPs and standard antibiotics against bacterial and fungal test cultures. Due to the smaller size and high concentration of AgNPs, it is worth using formulated AgNPs to control and kill pathogenic bacteria and fungi. The developed AgNPs from the GR-CHA-4 strain exhibited the highest inhibitory activity against *P. aeruginosa* (8 µg/mL), followed by *E. coli* and *A. niger* (32 µg/mL), then against *C. albicans* (64 µg/mL). *B. subtilis* (128 µg/mL) and *S. aureus* (256 µg/mL) were much less sensitive to AgNPs.

#### 2.3.13. Synergistic Effect of AgNPs 

The strategy of combining metal NPs with conventional antibiotics is a good alternative to combat growing bacterial resistance. The synergistic effect of NPs was studied against bacterial and fungal test cultures, and the FIC index and their interpretation values are shown in Table 2.

#### 2.3.14. AgNP Activity against MRSA

The rise of multi-drug resistance has necessitated the creation of a novel category of antimicrobial drugs. The biosynthesized AgNPs showed a good amount of activity against MRSA, with zone inhibition of 16 mm and 12 mm against MRSA 2 and MRSA 7, respectively.

#### 2.3.15. Anti-Biofilm Activity of AgNPs Synthesized with *S. tendae*

Generation of the sessile communities and their inborn multi-drug resistance is the basis of several chronic and relentless bacterial infections [54]. The anti-biofilm efficacy of AgNPs was evaluated against *P. aeruginosa* and *S. pneumoniae* by applying a crystal violet assay (Table 3).

#### 2.3.16. Anti-Oxidant Activity of AgNPs

An incremental rise in the percentage of free radical scavenging activity was noted (40.14, 52.12, 64.23, and 80.44%) as the quantity of AgNPs increased from 25 µg/mL to 200 µg/mL (Figure 8). 

#### 2.3.17. Preparation of Anti-Microbial Cotton

The AgNP-treated cotton effectively suppressed the growth of *Bacillus subtilis* and *E. coli*, resulting in the formation of ZOI of 15 mm and 12 mm, respectively.

## 3. Discussion

The change in color from pale yellow to dark brown was noticed after 4 to 24 h of incubation, as shown in Figure 9a,b. The color change suggested that actinomycete played a significant role in the development of AgNPs. Similar observations have been reported for *Bacillus licheniformis* [55], *P. aeruginosa* [56], *E. coli* [57], and *Bacillus megaterium* [58]. The mechanism of formation of AgNPs by the actinomycetes is well described in the literature. The actinomycetes, being metal-tolerant, take the Ag^+^ ions from the aqueous medium into the cell and cytoplasm (Figure 9b). The Ag^+^ ions are further reduced by the various biological molecules present inside the actinomycetes. Further, these reduced AgNPs are capped by biological molecules and released into the medium if the nature of the actinomycetes is extracellular, and the AgNPs have to be extracted by applying sonication and another approach to extract the AgNPs inside actinomycetes. Alkaliphilic actinomycetes *Nocardiopsis valliformis* from an alkaline Lonar crater were reported for the extracellular synthesis of AgNPs by Rathod and their group [59]. Extracellular biogenic synthesis of AgNPs was also found in the actinomycete isolated from marine sediment [60]. From the two sites of ROK, three potential actinomycetes were screened out for the synthesis of AgNPs. The three strains were coded as GR-CHA-4, GR-ADE-3, and GR-CHA-7, where the GR stands for Greater Rann of Kutch, CHA stands for Charanka Solar Park, and ADE stands for Adesar region. GR-CHA-4 and GR-CHA-7 were isolated from the Charanka solar park region, while GR-ADE-3 was isolated from the Adesar region of ROK. Out of all the three potential organisms, the AgNPs synthesized by GR-CHA-4 were of the desired shape and property. 

So, the synthesis of AgNPs from GR-CHA-4 was selected for further optimization based on spectral analysis (UV-Vis) of AgNPs, reaction time, and color change. Physico-chemical parameters such as growth medium, pH, temperature, and precursor (AgNO_3_) concentration were optimized for AgNP synthesis. The narrow surface plasmon peak obtained at 428 nm indicated the lower aggregation of particles. The SPR of NPs is affected by the morphology, concentration, and interparticle distance, which indicates its efficiency in monitoring the aggregation of NPs [61].

The results obtained for the optimization of the media for AgNPs proved that differences in the medium components affect the production of secondary metabolites and ultimately influence nanoparticle production using the isolates. Similar reports on MGYP for the biosynthesis of various NPs have been found [62]. So, there is a need to optimize the medium components using statistical design [63].

The results obtained for the synthesis of AgNPs under normal, alkaline, saline, and saline + alkaline conditions are discussed here. The results showed that the NaCl concentration interferes with the formation of NPs. The growth of NPs due to nucleation is not possible in the presence of salt, so 5% NaCl interferes with the AgNP formation. Our results are in agreement with the earlier report which showed silver nanoparticle synthesis in MGYP at pH 7 [52]. Waghmare et al. (2014) also showed that neutral pH is optimum for AuNP biosynthesis using *Streptomyces hygroscopicus* [64]. In contrast, Skladanowski et al. (2017) synthesized gold and AgNPs from *Streptomyces* sp. strain NH21 by growing in a medium having pH 5.5 [65].

A vast literature exists favoring the influence of a range of substrate concentrations on nanoparticle synthesis. Most of the investigations reported the use of 1 mM AgNO_3_ for AgNP synthesis from actinomycetes [66,67].

Thirumurugan et al. (2020) documented the synthesis of AgNPs from actinobacteria at a very low concentration [68]. Giri et al. found a very high concentration of AgNO_3_ as the optimum concentration to support nanoparticle synthesis [69]. AgNP synthesis using a high concentration of parental compounds (i.e., AgNO_3_) is beneficial in biomedical sector applications. The concentrated AgNPs can be diluted for the applicable formulation, so it is desirable to formulate AgNPs using a high concentration of AgNO_3_. The important parameter that influences the size and shape of NPs is the pH of the reaction mixture. This factor can alter the charge of secondary metabolites and bio-molecules involved in the formation of NPs, which may affect their stability and capping properties [70,71].

The above-mentioned results are from the recent study conducted on the synthesis of AgNPs from actinobacteria. Their findings showed the synthesis of comparatively smaller NPs with highly negative ZP (−26.2) values at pH 7. Contrary to these reports, acidic and alkaline pH also proved to support the biosynthesis of NPs [72]. The pH from acidic (pH 4.0) to neutral (pH 7.0) led to the formation of smaller and spherical NPs. Acidic or alkaline pH favors the formation of irregularly shaped NPs with agglomeration [73]. The shape of AgNPs at pH 7.0 is usually spherical, but acidic pH causes the formation of hexagonal or irregularly shaped NPs [74]. So, neutral pH is the best for the synthesis of stable and spherical NPs.

From the results obtained for the effect of temperature on AgNP synthesis, it was found that 60 °C was considered as an optimum temperature for AgNP synthesis. Beyond 60 °C, synthesis occurred, but at a slower rate with a lesser yield. Similar observations are reported on the synthesis of NPs in the presence of high temperatures. In one such investigation, investigators biosynthesized AgNPs in a boiling water bath at a much higher rate, where the investigators hypothesized the role of thermo-stable compounds in nanoparticle formation [72]. Similarly, an increase in yield concerning an increase in temperature was observed by Kiran et al., where the investigators found the synthesis of AgNPs at a temperature range of 40–100 °C, with maximum absorbance at 100 °C. The rate of reaction and kinetic energy in the reaction at high temperatures are increased, so rapid formation is found [75]. 

The Particle Size Analysis (PSA) and zeta potential (ZP) of the *S. tendae*-mediated synthesized AgNPs showed a resemblance to the results reported by Skladanowski et al. [65]. The small-sized spherical-shaped AgNPs (45 nm) from *Streptomyces* sp. have been reported. Furthermore, the biological synthesis of NPs with different shapes and sizes depends on the biological entity involved, reaction time, and metal ion concentration. For instance, *Streptomyces griseorubens* synthesized spherical-shaped AgNPs with 5–20 nm size [76]. Smaller particles show a larger surface area to volume ratio; hence, the size of NPs affects the bactericidal action of NPs.

The ZP value of the AgNPs was −20.4 mV (Figure 3b) and the present results were similar to the recent findings of Vijayabharathi et al. [77]. The author reported a negatively charged surface of AgNPs synthesized by *Streptomyces griseoplanus* with a ZP value of −20.4 mV. The negative ZP value could be assigned to the negatively charged functional groups present on the surface of NPs acquired from cellular metabolites [78]. Further, the narrow and centered ZP shows the stability of NPs in a colloidal suspension [79]. Furthermore, such negatively charged NPs are more suitable to use for drug antibiotic complex formation, which is valuable for biomedical applications.

FT-IR spectra of AgNPs exhibited bands in the region of 531 to 3328 cm^−1^. The vibration located at 3328 and 2111 cm^−1^ corresponds to the stretching vibration of the alcohol (ROH) group and C-O stretching mode [80,81], respectively. Furthermore, the FTIR spectrum also revealed a peak at 1635 cm^−1^, attributed to the C=O stretching vibrations in amide linkages (amide II) of protein present in cell-free supernatant [82,83]. Proteins contain several functional groups, like OH from amino acid side chains, like serine, threonine, and tyrosine, amide groups (from the peptide backbone), and C=O groups (from acidic side chains such as aspartic and glutamic acid) [84]. The microbially synthesized (*Streptomyces*) AgNPs cap and stabilize the developed NPs, preventing their aggregation. The capped protein molecules could involve amino acid residues with OH groups, along with H_2_O molecules associated with the protein structure. Previous FTIR analysis of AgNPs showed that proteins such as amide I (around 1650 cm^−1^) and amide II (around 1550 cm^−1^) are likely involved in the synthesis and stabilization of AgNPs [85].

Similarly, Muthusamy et al. reported AgNPs of *Streptomyces olivaceus* having six functional groups, C=C-H, ROH, C=N-OH, C-C, R-NH_2,_ and RCOOH, in the range of 455.20 to 3294.42 cm^−1^ [86]. The characteristic spectrum obtained indicates the presence of protein-like compounds attached to the synthesized NPs, confirming the role of metabolically produced proteins as capping agents during production and also interrupting the particle aggregation. Furthermore, the carbonyl groups are known for their silver-binding properties [87]. 

The sharp intensity peak in the XRD pattern indicates the crystalline nature and small crystallite size of the developed AgNPs. The typical five characteristic diffraction peaks indexed at the lattice plane were (111), (200), (231), (222), and (220). Apart from these, the recorded XRD pattern shows additional unassigned intense peaks that may be due to the formation of the crystalline bio-organic compounds/metalloproteins that are present in the culture broth. All the peaks were analyzed through the Joint Committee on Powder Diffraction Standards (JCPDS) and the pattern was in agreement with the standard values file number 89-3722. These data are in agreement with the previously conducted analysis by Składanowski et al., which documented the peaks at 2θ values (38.1°, 44.6°, 64.6°, 77.5°, 81.5°, and 115.0°) of crystalline AgNPs [65]. Likewise, Muthusamy et al. (2018) obtained the XRD pattern of synthesized AgNPs from *Streptomyces olivaceus* (MSU3), showing four intense peaks with 2θ values around 38.12°, 44.30°, 64.45°, and 77.41, with the lattice plane (111), (200), (220), and (311), respectively [86].

A team led by Manikprabhu and Lingappa (2013) performed the XRD characterization of AgNPs produced from *Streptomyces coelicolour* with the spectrum peak values of 75.41° and 40.21° [88]. Recently, Pallavi et al. documented the crystalline nature of AgNPs along with strong peaks in the range of 30 to 80° 2θ values from the *Streptomyces hirsutuS* strain SNPGA-8 [89]. 

The SEM images suggested the monodispersed nature of the AgNPs in the solution. Much literature is available on the use of SEM imaging in knowing the size and shape of bio-synthesized AgNPs. Spherical-shaped AgNPs with a diameter of 19.5–20.9 nm have been synthesized by Vijayabharathi et al. by using *Streptomyces griseoplanus* SAI-25 [77]. 

Recently, Ghany et al. synthesized monodispersed spherical AgNPs having an average size of 8.48 ± 1.72 nm and 9.67 ± 2.64 nm from actinomycetes *Glutamicibacter nicotianae* SNPRA1 and *Leucobacter aridicollis* SNPRA2, respectively [90]. Further, Nayka and their team also reported the formation of spherical and polydisperse AgNPs of size 32.40 nm from *Streptomyces* sp. NS-33 [91]. The majority of the studies show the formation of spherical-shaped AgNPs from actinomycetes isolates. However, several studies also reported the formation of rod-shaped AgNPs from *Nocardia mediterranei* [46,92].

The EDS clearly shows a high amount of Ag, which indicates the purity of the sample, while S and Cl are due to the media used for the halotolerant actinomycetes. Ca and Mg are present as an impurity, which is again from the media used for the actinomycetes. 

The AFM images were in close agreement with Zarina et al., who synthesized spherical and polydisperse AgNPs from 50 to 76 nm in size, and also observed the height and surface roughness properties of NPs using a 3D AFM micrograph. Likewise, the average size of AgNPs (68.13 nm) synthesized from actinomycetes was reported by Chauhan et al. [93]. Further, Sadhasivam et al. studied the spherical shape and homogenous nature of AgNPs [94]. Recently, AFM technique-based histogram analysis and surface roughness of the AgNPs were observed by Iniyan et al., who showed the presence of very small-sized NPs in the range of 6 to 7 nm [95].

From the antimicrobial activity of AgNPs, the MIC values showed that GNB was more sensitive to AgNPs than GPB. This might be due to the thinner cell wall of GNB than that of Gram-positive bacteria [96]. Similarly, Wypij et al. (2018) also reported the highest inhibitory action of AgNPs against *P. aeruginosa* (16 µg/mL), and *S. aureus* was the least sensitive among all the tested organisms (256 µg/mL) [73]. In contrast, Sivasankar et al. reported the MIC values of AgNPs synthesized from *Streptomyces violaceus* against Gram-positive bacteria (*S. aureus* and *B. subtilis*) and GNB (*E. coli* and *P. aeruginosa*) and found that *P. aeruginosa* (MIC 111.6 µg/mL) was the least sensitive of the tested bacteria [96]. 

Immanuel et al. (2018) demonstrated much lower MIC values in the range of 0.5 to 3.0 µg/mL against clinical pathogens. When tested against clinical pathogens, namely *E. coli*, *Enterococcus faecalis*, *Klebsiella pneumoniae*, *Streptococcus mutant*, and *S. pneumoniae,* the AgNPs fabricated by *Streptomyces olivaceus* exhibited MIC values of 1.25, 1.25, 2.5, 0.625, and 0.625 µg/mL, respectively [96]. 

The MIC value of anti-fungal activity reported by Manivasagan et al. against *A. niger* (16 µg/mL) and *C. albicans* (10 µg/mL) was lower than the MIC value [44] reported in our study, whereas our results are better than the latest study on the antifungal activity of actinomycetes conducted by Wypij et al. (2018), which showed MIC values of 64 µg/mL and 96 µg/mL against *C. albicans* [73]. Further, the anti-fungal potential of AgNPs synthesized from acidophilic actinomycetes was evaluated against a panel of fungal pathogens causing superficial mycoses, such as *C. albicans*, *C. tropicalis*, *Malassezia furfur*, and *Trichophyton rubrum,* with MIC values of 15, 16, 16, and 26 µg/mL, respectively.

Among the commercial antibiotics, tetracycline demonstrated the highest inhibitory activity against bacteria, followed by kanamycin and ampicillin. AgNPs exhibited higher inhibitory action against bacterial growth than ampicillin. In the case of kanamycin, the growth of GPB was inhibited at a much lower concentration than in the presence of AgNPs, while the GNB was found to be more sensitive to AgNPs than kanamycin. Similarly, a study reported higher bactericidal activity of the synthesized AgNPs in comparison to ampicillin, tetracycline, and kanamycin when tested against *Bacillus subtilis*, *E. coli*, *K. pneumoniae*, *P. aeruginosa*, *Proteus mirabilis*, *S. aureus*, and *Salmonella infantis* [97]. So, the synthesized AgNPs revealed more potentiality against GNB as compared to commercial antibiotics. The anti-fungal affectivity of AgNPs was also higher than fluconazole, except for amphotericin-B. 

As mentioned in earlier reports, the biocidal activity of AgNPs depends on the size and stability of particles. The findings are supported by various investigations that studied the size-dependent antimicrobial activity of AgNPs [73] and also demonstrated the microbicidal activity of AgNPs formulated by two different haloalkaliphilic actinobacteria and found that smaller-sized particles exhibited better antimicrobial activity [98].

Several mechanisms have been proposed for the biocidal activity of AgNPs. Several targets of AgNPs in the bacterial cell have been identified. (i) Disruption of the cell wall and cell membrane affects the regulation of the transport mechanism [99], (ii) increased permeability of the cell membrane leads to leakage of vital cell constituents and nutrients [100], and (iii) penetration of AgNPs inside the microorganisms is followed by interaction with biomolecules (lipids, proteins, and nucleic acid) [101].

The synergistic action of tetracycline and AgNPs was significant against both Gram-positive and Gram-negative test cultures. A combination of ampicillin and AgNPs demonstrated synergistic efficacy against GPB, and in Gram-negative bacteria, an indifferent effect was observed. Anti-fungal antibiotic fluconazole in combination with AgNPs also exhibited an indifferent effect against *C. albicans* and *A. niger*. The findings obtained in this study are consistent with the research conducted by Wypij et al. (2018) [73]. Anasane et al. conducted a study on the combined effectiveness of AgNPs, ketoconazole, and fluconazole against fungal infections [78]. Enhancement in the antibacterial activity of penicillin, streptomycin, ampicillin, and cephalexin against bacterial pathogens has been shown by various investigators [102,103]. The coadministration of AgNPs with antibiotics showed significantly increased antibacterial efficacy, while also resulting in reduced dosage requirements for both substances. Utilizing smaller concentrations of compounds can mitigate their toxicity while still achieving the desired outcome [104]. Additionally, investigators asserted that this combination effectively restores the ability of the medicine to kill microorganisms that have become resistant to it. For instance, *S. mutans* was found to be sensitive towards vancomycin treatment in the presence of AgNPs [105]. Likewise, the resistant strain of *P. mirabilis* showed sensitivity toward the same antibiotics in the presence of biologically synthesized AgNPs [106].

The results obtained for the MRSA activity of AgNPs were in close agreement with the previous results. In one of the investigations, the activity against MRSA by AgNPs synthesized by actinobacterium *Sinomonas mesophila* was reported. AgNPs synthesized by marine *Micromonospora* exhibited excellent bactericidal activity against a range of pathogens [88]. The efficacy reported was much higher than the standard antibiotic cefotaxime. Apart from silver, other NPs such as MgO and CuNPs from actinomycetes also exhibited their bactericidal action against MDR and MRSA strain [107,108].

The biofilm formation by *P. aeruginosa* showed more sensitivity towards AgNPs and standard antibiotic ampicillin compared to *S. pneumoniae*. In addition, the combinational approach showed remarkable inhibition of biofilm formation in both the tested cultures. About 98% inhibition was observed in the case of *P. aeruginosa,* whereas biofilm formation by *S. pneumoniae* decreased up to 83% in the presence of AgNPs and ampicillin. The results obtained by the biogenic AgNPs were shown to disrupt the biofilm growth mode of bacteria and fungi. Potential anti-biofilm activity (95% inhibition) of AgNPs synthesized by *B. licheniformis* was noticed in contradiction to *P. aeruginosa* and *S. epidermidis* [109].

The scavenging effect of vitamin C was 45.6% at 100 µg/mL concentration. Anti-oxidant activity of AgNPs is well-studied and well-known [110]. Due to good anti-oxidant activity, AgNPs are now commercially used in skin lotion and anti-aging creams. 

The application of AgNPs onto cotton fabric is attributed to the presence of a negative charge on the surface of the AgNPs. The positive functional group on the cotton fiber ties up with negatively charged AgNPs. The data obtained are very close to the work reported by Yan et al., [111]. Likewise, textile fabric loaded with copper NPs was also evaluated for antimicrobial efficiency against *S. aureus* and *E. coli* [112]. The utilization of ZnNPs has also been suggested in the production of antimicrobial packaging to enhance the durability of packaged materials [113,114].

## 4. Materials and Methods

### 4.1. Materials 

The collection of the soil samples was carried out from Greater Rann of Kutch, i.e., Charanka Solar Park and Adesar (Kutch, India), Starch Casein Agar (SCA) (HI-media Mumbai, India, Analytical grade), NaCl, and AgNO_3_ (98%, AR grade, Rankem, Mumbai, India), and double distilled water (ddw)_._ Microbial strains *Bacillus subtilis* MTCC 441, *Staphylococcus aureus* MTCC 737, *Escherichia coli* MTCC 1687, *Pseudomonas aeruginosa* MTCC 1688, *Candida albicans* MTCC 183, and *Aspergillus niger* MTCC 1344 were procured from Microbial Type Culture Collection Center (MTCC), Chandigarh, India. MRSA 2 and MRSA 7 were isolated from the laboratory with Whatman filter paper No 1 (Merck, Mumbai, India).

### 4.2. Methods 

#### 4.2.1. Sample Collection Site

The soil sample was collected from the greater Rann of Kutch from Kutch district of Gujarat, India, which is shown below in Figure 10. Samples were collected from two sites, one from Charanka Solar Park (latitude: 23.8952° N and longitude: 71.2261° E, Code: GR-CHA), and another from Adesar region (latitude: 23.5581° N and 70.9814° E), and code GR-ADE. 

#### 4.2.2. Enrichment and Isolation of Actinomycetes

The enrichment and isolation of actinomycetes were carried out in the following sequence order, as shown in Figure 11 [115]. 

#### 4.2.3. Effect of Salt Concentration on the Growth of Isolate

The effect of salt concentration on growth was studied by supplying the isolates with a range of NaCl concentrations of 0.0, 2.5, 5.0, and 10.0 (% *w*/*v*) on SCA plates. After 7 days of incubation, the tolerance limits of actinomycetes to NaCl were determined and analyzed comparatively. Incubation was carried out as per the optimum temperature observed in the earlier study [116].

#### 4.2.4. Effect of pH on the Growth of Isolates

The influence of pH on growth was demonstrated by using SCA plates having different pHs, from 5.0 to 9.0. The acidic pH was adjusted with 5% sterile CH_3_COOH solution, and the alkaline pH was adjusted with 20% sterile Na_2_CO_3_ solution after separate autoclaving. After 7 days of incubation, the tolerance limits of the isolate to pH were determined and analyzed comparatively. Salt and temperature were used for each isolate as per the results of the growth of isolates under normal, alkaline, saline, and saline + alkaline conditions. Growth response was also investigated with the stress-supplemented condition, i.e., alkaline, saline, and saline + alkaline. Four different combinations (Table 4) of pH and salt were used to study the haloalkaliphilic nature of the isolates. Control was kept without any stress [117].

#### 4.2.5. Screening for the Presence of Antagonistic Substances

Actinomycetes are excellent elaborators of pharmaceutical products and are well known for finding novel biologically active secondary metabolite [118,119]. The primary screening of antibiotics was conducted using the top agar method. Isolates were grown on SCA (5.0%, *w*/*v*, NaCl, pH 9.0) for one week. After incubation, the top agar was mixed with the test culture overlaid on the base agar and incubated for one day. The zone of inhibition (ZOI) surrounding the actinobacterial colony was observed, and results were recorded in mm. The screening of all the isolated strains for their antimicrobial activity against MTCC strains was performed. 

#### 4.2.6. Molecular Identification of the Actinomycete Isolates

The identification of the isolated actinomycete was carried out by using 16S rRNA gene sequencing where pure culture strain on SCA slants was outsourced (Chromous Biotech Pvt. Ltd., Bangalore, India). The 16s rRNA sequence alignment was carried out manually with the available nucleotide sequence retrieved from EzTaxon [120,121]. The isolate was identified using the EzTaxon database (Table 5). Tylosin and nikkomycin-like compounds have been reported from *Streptomyces tendae* [122].

#### 4.2.7. Biosynthesis of AgNPs

About 30 mL of malt, glucose, yeast peptone (MGYP) medium made up of malt and yeast extract each (0.3%), glucose (1.0%), and peptone (0.5%) (pH-7.0, NaCl 0.0% *w*/*v*) was inoculated from a 1-week-old culture grown on an SCA plate (pH-9.0; NaCl 5.0% *w*/*v*). The incubation of all the flasks was carried out along with shaking at 120 rpm at 30 °C for 72 h. About 10 mL of inoculum was inoculated into 50 mL MGYP (pH 7) after that incubation of the flasks was carried out at 30 °C along with shaking at 100 rpm and for 96 h. After incubation, the filtration of the biomass was carried out using Whatman filter paper No. 1, where the filtrate containing crude nitrate reductase (NR) was subsequently used for biosynthesis. The supernatant was exposed to 1 mM of AgNO_3_ aqueous solution in equal amounts and the mixture was left undisturbed under static conditions in the dark at 30 ℃, while the control had only AgNO_3_ aqueous solution and no actinomycete supernatant culture [123,124]. 

#### 4.2.8. Optimization of Physicochemical Parameters 

The isolated actinomycete was further optimized for extracellular nanoparticle synthesis using pH, temperature, AgNO_3_ concentration, medium composition, and under-stress conditions.

##### Medium Optimization for AgNP Synthesis

To reveal the effect of components of the media on the production of secondary metabolites involved in the formation of NPs, the isolate was grown in starch casein broth, ISP medium No 1, M9 (sodium acetate as a carbon source), casein glycerol broth, and MGYP medium. NaCl was not added in any medium and pH was set to 7.0. The biosynthesis of AgNPs was carried out according to the protocol described in 3.19.

##### Nanoparticle Synthesis under Normal, Alkaline, Saline, and Saline + Alkaline Conditions

To optimize the pH effect and concentration of NaCl on the ability of AgNP production of the selected isolate, all the isolated strains were grown in the presence of 4 different conditions, i.e., normal (0.0% *w/v* salt, 7.0 pH), alkaline (0.0% *w/v* salt, 9.0 pH), saline (5.0% *w/v* salt, 7.0 pH), and saline + alkaline (5.0% *w/v* salt, 9.0 pH). 

##### Effect of AgNO_3_ Concentration on AgNP Synthesis

The effect of AgNO_3_ on the formation of AgNPs was observed using 1 mM to 5 mM AgNO_3_ concentration at pH 7.0 and 30 °C. During incubation, the color change of the reaction solution was observed at different time intervals. Further confirmation was carried out based on UV–visible spectra.

##### Effect of pH on AgNP Synthesis

The influence of pH on the biosynthesis of NPs was studied in the presence of pH 5.0, 6.0, 7.0, 8.0, and 9.0. The reaction mixture was set to different pHs followed by incubation at 30 °C in the dark. During incubation, the color change of the reaction mixture was observed at different time intervals. Further confirmation was carried out based on UV–visible spectra.

##### Effect of Temperature on AgNP Synthesis

The relationship between temperature and nanoparticle formation was evaluated at different temperatures from 30 to 80 °C. The confirmation of the synthesis of AgNPs was revealed by the color change of the mixture (pale yellow to brown). Further confirmation was carried out by performing UV–visible spectroscopy.

#### 4.2.9. In Vitro Antimicrobial Activity of AgNPs

The antimicrobial activity of the synthesized AgNPs was assessed against *S. aureus* MTCC 737, *B. subtilis* MTCC 441, *E. coli* MTCC 1687, *P. aeruginosa* MTCC 1688, *C. albicans* MTCC 183, and *A. niger* MTCC 1344. To determine the MIC of AgNPs, the microtiter broth dilution technique was used as per the guidelines given by the Clinical Laboratory Standards Institute (CLSI) using 96-well ELISA microtiter plates. Muller–Hinton (MH) broth and Sabouraud dextrose (SD) broth were utilized for growing bacteria and fungi, respectively. Test cultures were grown in respective mediums and diluted to 0.4 McFarland standards for MIC testing. To each well (capacity 300 µL), 185 µL medium and 10 µL of AgNPs suspended in distilled water were added. Contents in the well were mixed properly and the following concentrations of AgNPs were prepared (1024, 512, 256, 128, 64, 32, 16, 8, 4, and 2 µg/mL) using a two-fold dilution method, which was followed by the addition of 5 µL of test culture to each well. The positive control (broth + microbial inoculums without AgNPs) and negative control (sterile un-inoculated broth) were also kept along with the experimental sets. Tetracycline, kanamycin, and ampicillin were used as standard anti-bacterial antibiotics, while amphotericin-B and fluconazole were used as anti-fungal antibiotics. The plates were incubated at 37 °C for bacteria and 28 °C for fungi for 2–3 days. After incubation, plates were read for absorbance at 600 nm using a spectrophotometer. MIC was estimated from the lowest concentration that inhibited the 100% growth of the test culture by comparing it with the negative control. The 5 µL aliquot of the sample showing complete inhibition of microbial growth was spotted on the respective medium for the estimation of the Minimum Bactericidal Concentration (MBC) of AgNPs against bacterial and fungal test cultures. The lowest concentration without bacterial colony on the agar plate was considered as MBC of AgNPs. 

#### 4.2.10. Synergistic Activity of AgNPs with Commercial Antibiotics

To evaluate the synergistic role of AgNPs with commercial antibiotics, the MIC value of individual agents (AgNPs and antibiotics) was combined in a single well and serially diluted to achieve the 1/2 MIC, 1/4, 1/8, and 1/16 MIC concentrations in the subsequent wells. Tetracycline and ampicillin were taken as anti-bacterial antibiotics and amphotericin B was used as an anti-fungal agent. The microtiter plate well with no turbidity was used to calculate the fractional inhibitory concentration (FIC) index. To calculate the FIC index, Equation (1) was used: (1)$$\mathrm{FIC}=\frac{\mathrm{MIC}\;\mathrm{of}\;\mathrm{AgNPs}\;\mathrm{in}\;\mathrm{combination}\;\mathrm{with}\;\mathrm{antibiotic}}{\mathrm{MIC}\;\mathrm{of}\;\mathrm{AgNPs}\;\mathrm{alone}}+\frac{\mathrm{MIC}\;\mathrm{of}\;\mathrm{antibiotic}\;\mathrm{in}\;\mathrm{combination}\;\mathrm{with}\;\mathrm{AgNPs}}{\mathrm{MIC}\;\mathrm{of}\;\mathrm{antibiotic}\;\mathrm{alone}}$$

#### 4.2.11. Activity against MRSA

MRSA strains isolated from the clinical samples were used in the study. The activity of the synthesized AgNPs against MRSA was evaluated by utilizing the agar well diffusion method with some modifications [125]. Briefly, MRSA strain grown in Mueller–Hinton broth having turbidity equal to 0.4 McFarland was spread heavily onto Mueller–Hinton plates. With the help of the sterile borer, wells were formed on the plate. Then, 50 μL of AgNPs sample was added into their respective cups. Plates were incubated at 20 °C for 30 min to facilitate the diffusion followed by incubation at 37 °C overnight. The next day, the ZOI was measured in mm and results were recorded. The isolate inhibited the growth of the MRSA strain with a ZOI of 12 mm against MRSA-7.

Metabolites against MRSA have been reported from saline desert isolates by Damavandi et al. [126]. Recently, inhibition of the staphyloxanthin-based activity against MRSA has been demonstrated from Streptomyces by a team led by Charousova [127].

#### 4.2.12. Anti-Biofilm Activity

Inhibition of biofilm formation by *P. aeruginosa* and *S. pneumoniae* was investigated in the presence of AgNPs alone and in combination with the standard antibiotic ampicillin using the tissue culture plate method [128,129]. Individual wells of the plates were poured, with the 190 µL of testing culture having 1 × 10^6^/mL concentration, and the incubation of the plates was carried out at 37 °C for 6 h, which was followed by washing with Phosphate Buffer Saline (PBS) to eliminate planktonic forms of bacteria. The biofilm developed on the surface of the well was fixed using sodium acetate (2%) followed by staining with crystal violet (0.1% *w*/*v*) and rinsing off the excess strain, followed by washing with Milli-Q water and air drying. After drying, wells were filled with 95% ethanol to make a suspension of adherent bacteria of the biofilm. The measurement of optical density (O.D) was carried out by an ELISA plate reader at 595 nm. The inhibition percentage of biofilm development was estimated by using Equation (2) [130].
(2)$$1=\frac{OD_{595}\;\mathrm{of}\;\mathrm{cells}\;\mathrm{treated}\;\mathrm{with}\;\mathrm{AgNPs}\;}{OD_{595}\;\mathrm{of}\;\text{non-treated}\;\mathrm{control}}\times 100$$

#### 4.2.13. Antioxidant Activity of Synthesized AgNPs

Antioxidants protect the cell by neutralizing the free radicals and thereby play a significant role in biomedical applications [131]. About 1 mL of different concentrations of AgNPs (25, 50, 100, and 200 µg/mL) was added to 0.5 mL 1, 1-diphenyl-2-picrylhydrazyl (DPPH) (1 mM). The mixture was placed under dark conditions for half an hour followed by UV-Vis measurement at 517 nm. A control containing DPPH without AgNPs was used, and vitamin C (ascorbic acid) was considered as standard. Lower absorbance of the reaction mixture suggests a higher DPPH radical scavenging activity. The percentage of scavenging activity was estimated by applying Equation (3).
(3)$$\%\;\mathrm{of}\;\mathrm{RSA}=\frac{\mathrm{Abs}\;\mathrm{control}\;-\;\mathrm{Abs}\;\mathrm{sample}}{\mathrm{Abs}\;\mathrm{control}}\times 100$$
where: RSA = Radical scavenging activity;Abs control = absorbance of DPPH radical + ethanol;Abs sample = absorbance of DPPH radical + AgNPs.

#### 4.2.14. Biosynthesis of AgNPs on Cotton Fabrics

The antimicrobial activity of AgNP-coated cotton fabrics was evaluated by utilizing the disc diffusion process. Before being used, cotton textiles were cleaned, sterilized, and dried, according to Shilpa et al. (2023) [132]. AgNPs were applied onto the surface of cotton cloths measuring 1 cm × 1 cm and then left to dry in the air. The effectiveness of silver nanoparticle (AgNP)-treated cotton fabrics in killing bacteria was assessed by examining their antibacterial potential against both the Gram-positive bacteria (GPB) and the Gram-negative bacteria (GNB) on Mueller–Hinton Agar (MHA) plates at various time intervals. A control was established using a cell-free supernatant that did not contain AgNPs. ZOI was detected on the plates after they had been incubated for one day at 37 °C [133].

### 4.3. Characterization of the Synthesized AgNPs

All the optimized conditions were put together in one experiment, and the AgNPs developed were analyzed using the following instruments.

#### 4.3.1. UV-Vis Spectral Analysis

The synthesis of the AgNPs in the supernatant was confirmed by withdrawing 3 mL of the colloidal sample at regular time intervals. Surface Plasmon Resonance (SPR) spectra were measured in the range of 250 nm to 750 nm with a 10 mm path length quartz cuvette using a UV-Vis spectrophotometer (Shimadzu, Tokyo, Japan, UV-1800). 

#### 4.3.2. Nanoparticle Tracking Analysis (NTA) and Zeta Potential of AgNPs

The particle size distribution of the synthesized AgNPs was measured using NTA, Version 2.3 Build 0034. The sample was prepared by adding 1–2 mg powder sample in a 10 mL Milli-Q water in a tube followed by sonication for 10 min to obtain finely dispersed particles. The confirmation of the stability of AgNPs was carried out using zeta potential (ZP) analysis (ZS 90; Malvern Instruments Ltd., Malvern, UK).

#### 4.3.3. Fourier Transform Infra-Red (FT-IR) Analysis

The analysis of functional groups on the surface of AgNPs was conducted using FT-IR. The FT-IR spectroscopic investigation was conducted using a Tensor-27 instrument (Bruker, Ettlingen, Germany) with a scan range of 400–4000 cm^−1^ at a scanning speed of 2 mm/S. The thoroughly desiccated AgNPs powders were processed with high-quality KBr for consolidation in a ratio of 1:50 for the analysis.

#### 4.3.4. SEM

The surface features of the synthesized AgNPs were analyzed using a SEM-Zeiss Evo MA 15, (Cambridge, UK), instrument, operating at an accelerating voltage of 20 kV. The AgNP sample was placed on the carbon tape, which in turn was fixed on an aluminum stub, and micrographs were taken. The elemental analysis of the sample was carried out using EDS attached along with the SEM. The elemental analysis was carried out with the help of an Oxford Instruments (Abingdon, Oxfordshire, UK) EDS analyzer. 

#### 4.3.5. X-ray Diffraction (XRD) Analysis

The fingerprint characterization of crystalline metallic silver and its structure determination was performed by XRD by using a Rigaku D/max 2550 X-ray diffractometer (Tokyo, Japan). The measurement of the powder sample was carried out in the range of 2θ, ranging from 20° to 70°, with a 0.05°/S scan rate. 

#### 4.3.6. AFM

The 3D and 2D images were taken by AFM, by using Multiview 1000 Nanonics, (Malcha Jerusalem, Israel), instrument. The dispersed mixture of AgNPs prepared above during UV was spread on a fresh mica sheet with the help of a micropipette. Further, the slide was allowed to dry, followed by washing with Milli-Q water to remove unattached particles. 

## 5. Conclusions 

The present study successfully explored the potential of the halotolerant actinomycetes, *Streptomyces tendae*, isolated from the Rann of Kutch, India, for the green synthesis of AgNPs in a single step. The supernatant of *S. tendae* possesses several secondary metabolites and extracellular enzymes that are responsible for the reduction of silver salt into NPs. The confirmation of the synthesized AgNPs by analytical instruments showed the formation of 20–90 nm by SEM and an average particle size of 50 nm by PSA. The size, shape, and yield of AgNPs by the *Streptomyces* spp. are influenced by various parameters like types of microorganisms used, pH, temperature, nutrient medium, etc. The highest antimicrobial activity of the AgNPs was observed against *Pseudomonas aeruginosa*, and they also showed anti-MRSA activity. The capability of the AgNPs in inhibiting biofilm development and antioxidant activity suggests the synthesized AgNPs as having a potential therapeutic role. The synergistic effect of AgNPs with antibiotics like penicillin demonstrated a significant reduction in the formation of biofilm, suggesting their potential application in increasing antibiotic efficacy. The application of AgNPs in the preparation of antimicrobial surgical cotton showcases their practical application in the biomedical field. In brief, the investigation combines microbiology, nanotechnology, and biomedical sciences to construct eco-friendly antimicrobial compounds from halotolerant actinomycetes to address multidrug-resistant pathogens. Finally, this investigation showcases the various applications of AgNPs and lays the groundwork for in-depth investigations into the biomedical potential of halotolerant actinomycetes from desert environments.

## 6. Future Prospects and Limitations 

The present study could be further used to enhance antimicrobial therapies, by using novel antibiotic combinations (synergism) and targeted drug delivery. Moreover, the applications of the developed AgNPs will be analyzed for the in vivo studies. In addition to this, the developed AgNPs could be applied for wound healing, cancer therapy, and other biomedical applications. By addressing all these limitations and expanding the scope of the applications, the present investigation and the developed AgNPs has the potential to contribute significantly to the field of biomedicine and the environment.

## Figures and Tables

**Figure 1 pharmaceuticals-17-00743-f001:**
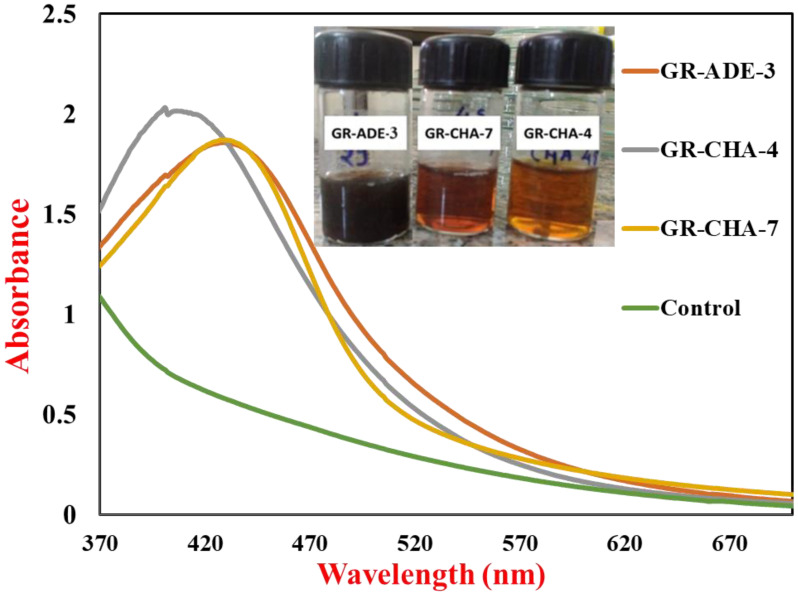
UV-Vis spectra of the AgNPs synthesized by three isolates along with the control reaction.

**Figure 2 pharmaceuticals-17-00743-f002:**
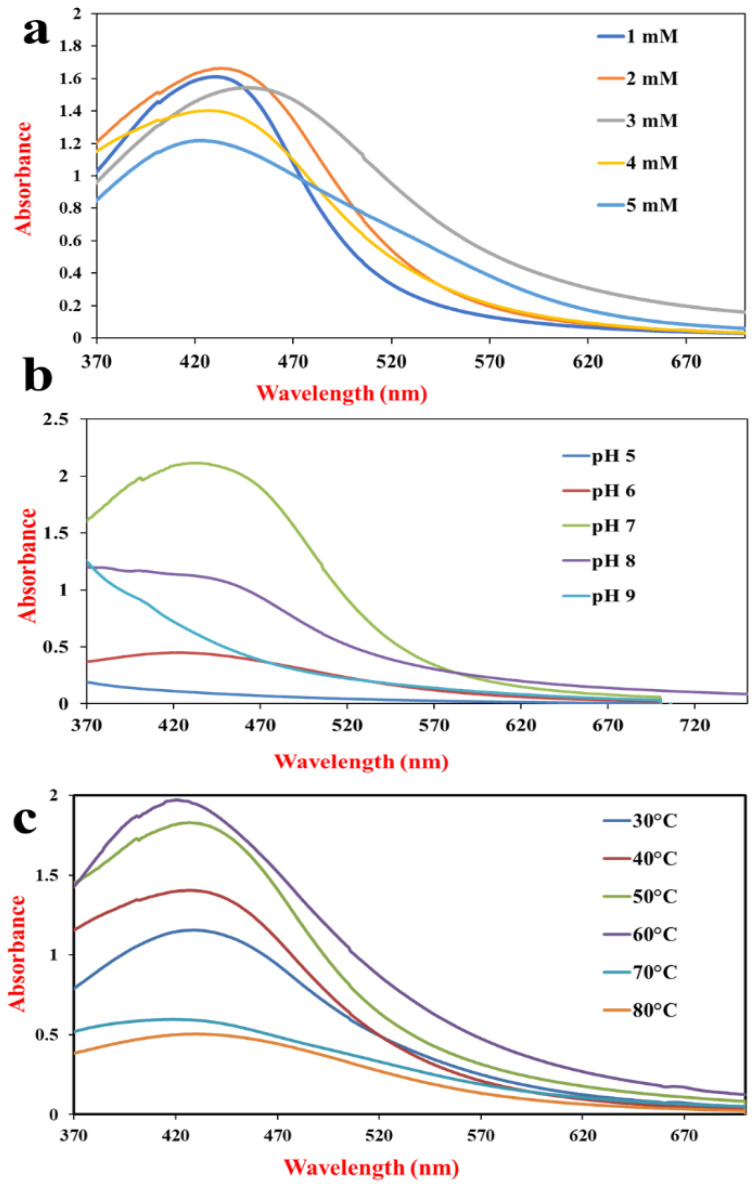
Effect of (**a**) AgNO_3_ concentration, (**b**) pH, and (**c**) temperature on AgNPs synthesis by *S. tendae* GR-CHA-4.

**Figure 3 pharmaceuticals-17-00743-f003:**
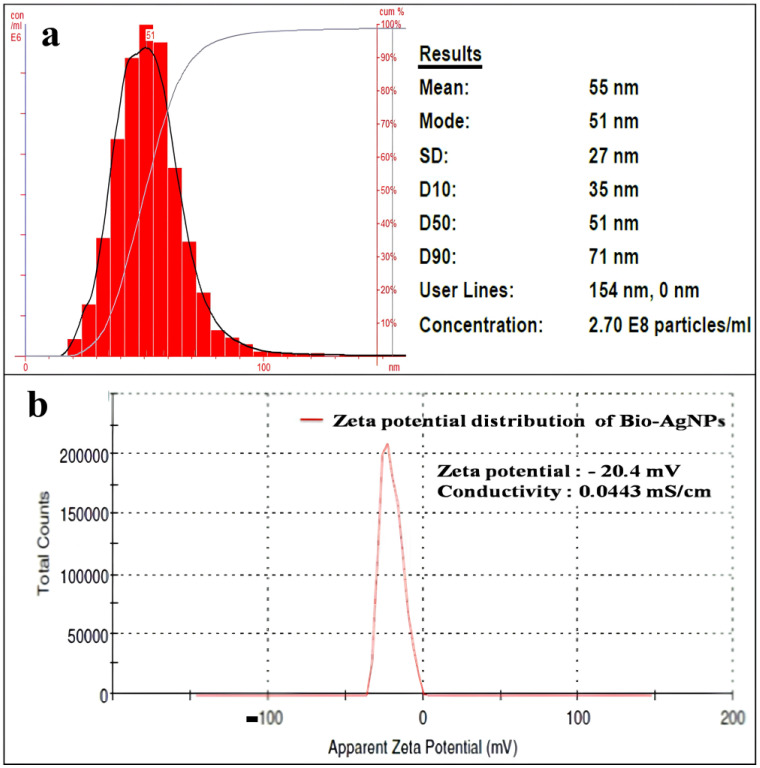
Particle size distribution analysis of AgNPs synthesized by *S. tendae* GR-CHA-4, and (**a**) zeta potential (**b**).

**Figure 4 pharmaceuticals-17-00743-f004:**
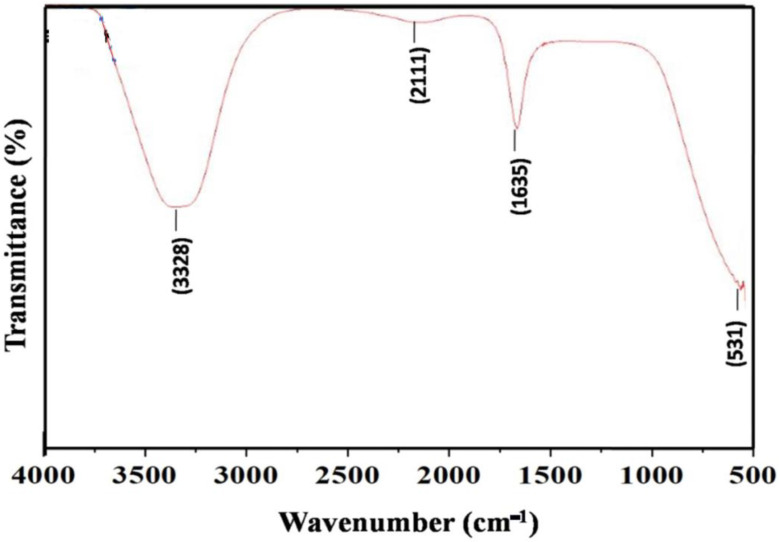
FTIR spectroscopy of AgNPs synthesized by *S. tendae* GR-CHA-4.

**Figure 5 pharmaceuticals-17-00743-f005:**
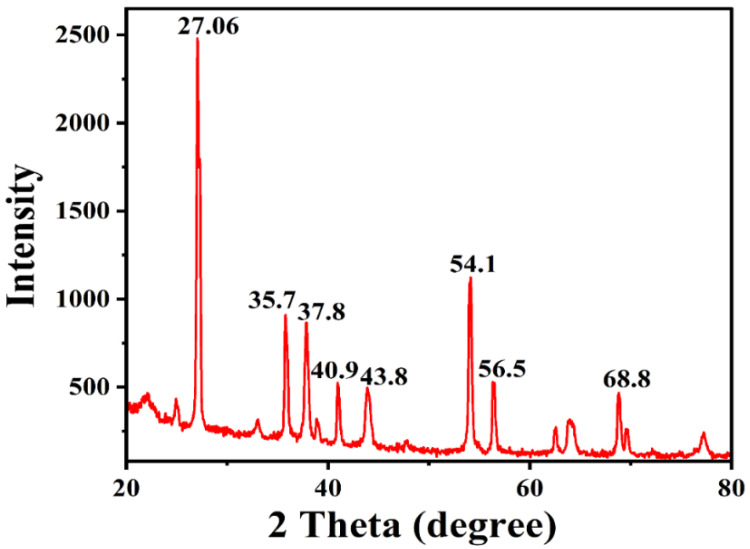
XRD pattern of AgNPs synthesized by *S. tendae* GR-CHA-4.

**Figure 6 pharmaceuticals-17-00743-f006:**
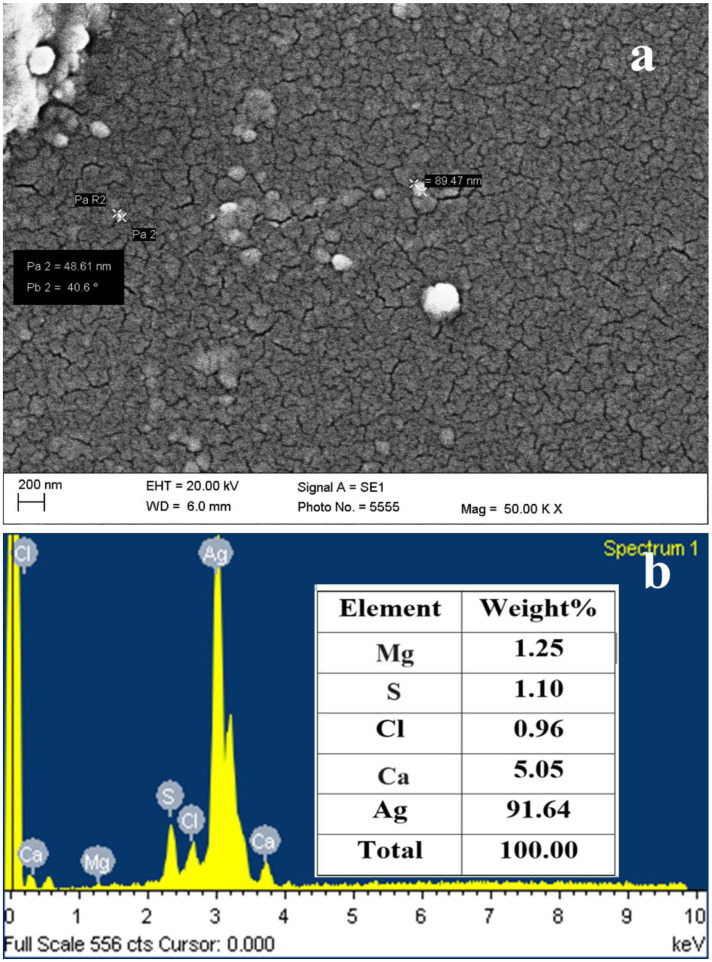
SEM micrograph (**a**) and EDS spectra (**b**) of AgNPs synthesized by *S. tendae* GR-CHA-4.

**Figure 7 pharmaceuticals-17-00743-f007:**
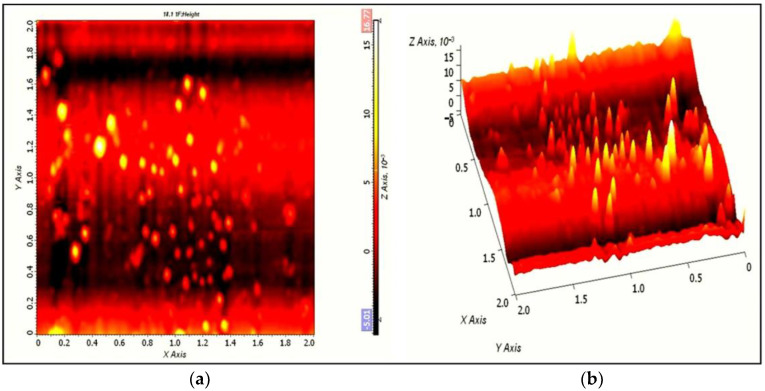
Two-dimensional (**a**) and three-dimensional (**b**) AFM images of AgNPs synthesized by *S. tendae* GR-CHA-4.

**Figure 8 pharmaceuticals-17-00743-f008:**
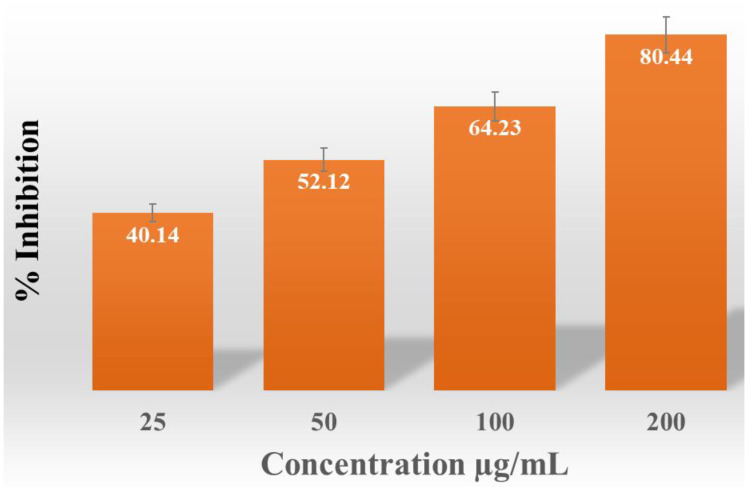
Anti-oxidant activity of AgNPs produced from *S. tendae*.

**Figure 9 pharmaceuticals-17-00743-f009:**
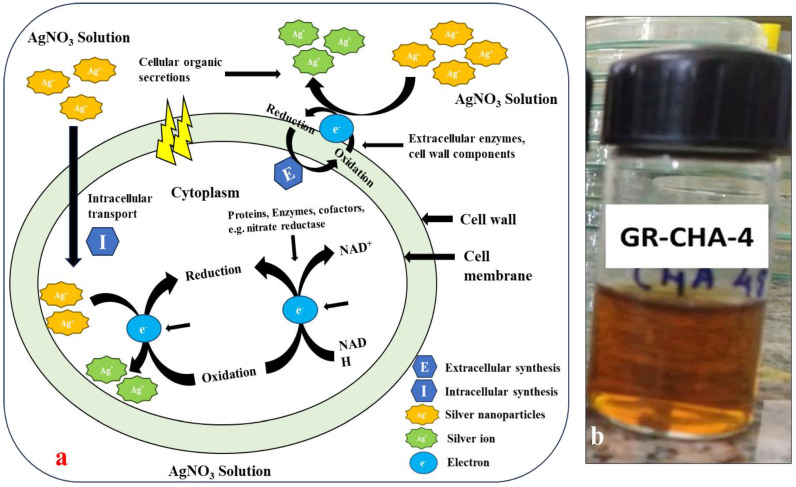
Mechanism of formation of AgNPs by actinomycetes (**a**) and color change for AgNP synthesis using actinomycete isolates (**b**).

**Figure 10 pharmaceuticals-17-00743-f010:**
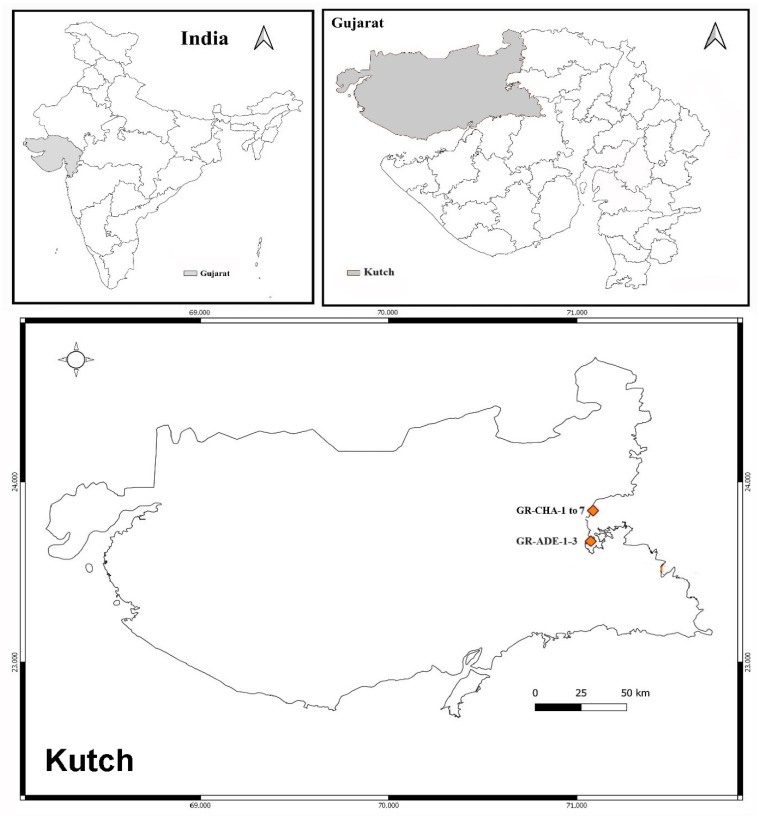
Soil sample collection site from Kutch: Greater Rann of Kutch.

**Figure 11 pharmaceuticals-17-00743-f011:**
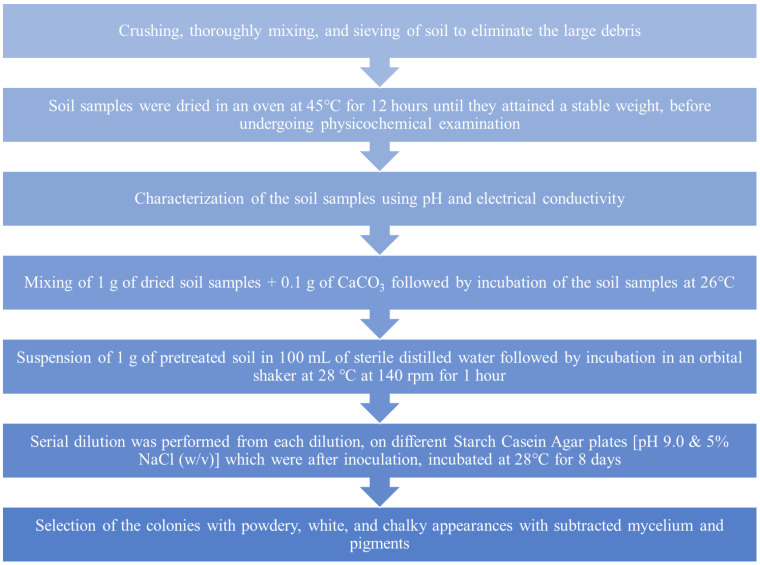
Steps involved in the enrichment and isolation of actinomycetes from desert soil.

**Table 1 pharmaceuticals-17-00743-t001:** Antimicrobial activity of AgNPs (MIC and MBC).

S. No.	Tested Microorganism	MIC of AgNPs (µg/mL)	MBC of AgNP (µg/mL)	MIC of Antibiotic (µg/mL)		
				AMP	K	TE
1	*B. subtilis* MTCC 441	128	128	128	8	4
2	*P. aeruginosa* MTCC 1688	8	16	512	12	64
3	*S. aureus* MTCC 737	256	256	128	3	64
4	*E. coli* MTCC 1687	32	64	512	128	64
				AMB	FLU	
5	*C. albicans* MTCC 183	32	64	8	256	
6	*A. niger* MTCC 1344	64	64	4	512	

AMP—ampicillin, K—kanamycin, TE—tetracycline, AMB—amphotericin B, FLU—fluconazole.

**Table 2 pharmaceuticals-17-00743-t002:** Synergistic effect of AgNPs with conventional antibiotics.

S. No.	Tested Microorganism	AgNPs + Ampicillin (µg/mL)	FIC	AgNPs +Tetracycline (µg/mL)	FIC	AgNPs+ Amphotericin B (µg/mL)	FIC
1	*Bacillus subtilis* (MTCC 441)	64 + 64	1	16 + 0.5	0.25		
2	*P. aeruginosa* (MTCC 1688)	2 + 128	0.5	1 + 8	0.25		
3	*S. aureus* (MTCC 737)	128 + 64	1	64 + 16	0.5		
4	*E. coli* (MTCC 1687)	8 + 128	0.5	4 + 8	0.375		
5	*C. albicans* (MTCC 183)					16 + 4	1
6	*A. niger* (MTCC 1344)					32 + 2	1
**FIC index**			**Interpretation**				
≤0.5			Synergistic				
>0.5–1.0			Additive or non-synergistic				
1.0–4.0			In different				
>4			Antagonistic				

**Table 3 pharmaceuticals-17-00743-t003:** Anti-biofilm activity of AgNPs.

Tested Microbes	Percentage Inhibition						
	AgNPs Dose (µg/mL)					AMP	AgNPs + AMP
	**10**	**20**	**30**	**40**	**50**	**50**	**30 µg/mL+ 30 µg/mL**
*P. aeruginosa*	15	22	30	46	60	35	98
*S. pneumoniae*	12	20	25	32	44	28	83

**Table 4 pharmaceuticals-17-00743-t004:** Combination of stress conditions.

Condition	pH	NaCl (%, *w*/*v*)
Normal	7.0	0.0
Alkaline	9.0	0.0
Saline	7.0	5.0
Saline + alkaline	9.0	5.0

**Table 5 pharmaceuticals-17-00743-t005:** Identification of isolates using the EzTaxon database.

S. No.	EzTaxon Database Suggested Species	Selected Species	Hit Strain Name	Similarity (%)
1	*S. tendae*	*S. tendae*	ATCC 19812(T)	98.46
2	*Streptomyces violaceorubidus*		LMG 20319(T)	98.46
3	*Streptomyces gougerotii*		NBRC 3198(T)	98.46

## Data Availability

Data is contained within the article.

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
