# Peer review of "Exploring the Potential of Halotolerant Actinomycetes from Rann of Kutch, India: A Study on the Synthesis, Characterization, and Biomedical Applications of Silver Nanoparticles"

_pharmaceuticals, 2024, doi:10.3390/ph17060743_

Round 1
Reviewer 1 Report
Comments and Suggestions for Authors
The manuscript describes the synthesis and characterization of silver nanoparticles (AgNPs) using Streptomyces tendae isolated from Rann of Kutch, India. After carefully reading this work, I have some observations:
1. There are some typos and grammatical mistakes. I suggest rereading the text.
2. Add the significance of the study to the abstract.
3. Lines 20 and 21: “UV-Vis spectrophotometer” is repeated in the same sentence.
4. The introduction is too long. Consider consolidating it.
5. Use more concise and precise language to convey the main ideas of the article.
6. Add the coordinates of the collection sites.
7. Figure 1: Avoid using colors for the schematic diagram or flow chart. Grayscale would be easier to read.
8. Tables 1-4: Please follow the MDPI instructions for tables.
9. Equations are inconsistent with the text.
10. Page 9 onward: the subsection numbering is wrong. Please fix it.
11. Photos are of low quality.
12. The quality of Figures 3, 4, 5, and 8 is not sufficient for a publication.
13. The font size is too big in almost all the figures. Please fix it.
14. The discussion is too long. Consider consolidating your arguments in a concise way.
15. Avoid repeating the same information multiple times and focus on the most important findings.
Comments on the Quality of English LanguageThe work would benefit from close editing.
Author Response
The manuscript describes the synthesis and characterization of silver nanoparticles (AgNPs) using Streptomyces tendae isolated from Rann of Kutch, India. After carefully reading this work, I have some observations:
- There are some typos and grammatical mistakes. I suggest rereading the text.
A/R: Thank you for pointing out this error. The authors have now thoroughly rectified all such mistakes in the revised manuscript as suggested by the reviewer.
- Add the significance of the study to the abstract.
A/R: Thank you for this valuable suggestion. The authors have now added the significance of the study in the revised abstract as suggested by the reviewer.
- Lines 20 and 21: “UV-Vis spectrophotometer” is repeated in the same sentence.
A/R: Thank you for pointing out this error. The authors have now rectified the mistake in the revised manuscript as suggested by the reviewer.
- The introduction is too long. Consider consolidating it.
A/R: Thank you for this valuable suggestion. The authors have now shortened the introduction section by keeping only the relevant information in the revised manuscript as suggested by the reviewer.
- Use more concise and precise language to convey the main ideas of the article.
A/R: Thank you for this valuable suggestion. The authors have modified the article accordingly.
- Add the coordinates of the collection sites.
A/R: Thank you for pointing out this error. The authors have now added the coordinates of the collection sites in the revised manuscript as suggested by the reviewer.
- Figure 1: Avoid using colors for the schematic diagram or flow chart. Grayscale would be easier to read.
A/R: Thank you for this valuable suggestion. The authors have now modified the flow chart as per the suggestion of the reviewer.
- Tables 1-4: Please follow the MDPI instructions for tables.
A/R: Thank you for this valuable suggestion. The authors have now modified tables 1-4 as per the instructions of the MDPI journal.
- Equations are inconsistent with the text.
A/R: Thank you for pointing out this error. The authors have now rectified the issues related to the equations in the revised manuscript as suggested by the reviewer.
- Page 9 onward: the subsection numbering is wrong. Please fix it.
A/R: Thank you for pointing out this error. The authors have now rectified the error in the revised manuscript as suggested by the reviewer.
- Photos are of low quality.
A/R: Thank you for this valuable comment. The authors have now provided images of higher resolution.
- The quality of Figures 3, 4, 5, and 8 is not sufficient for a publication.
A/R: Thank you for this valuable comment. The authors have now improved the quality of Figures 3, 4, 5, and 8 to the highest possible level. Some of the figures were provided as such by a third party, for which we don’t have raw data and are unable to process it.
- The font size is too big in almost all the figures. Please fix it.
A/R: Thank you for this valuable suggestion. The authors have now kept a uniform font size for all the figures in the revised manuscript as suggested by the reviewer.
- The discussion is too long. Consider consolidating your arguments in a concise way.
A/R: Thank you for this valuable suggestion. The authors have now reduced the discussion length by keeping the most relevant information in the revised manuscript as suggested by the reviewer.
- Avoid repeating the same information multiple times and focus on the most important findings.
A/R: Thank you for pointing out this mistake. The authors have now removed all such repetitive information from the revised manuscript as suggested by the reviewer.
- Comments on the Quality of English Language. The work would benefit from close editing.
A/R: Thank you for this valuable suggestion. The authors have now thoroughly edited the manuscript in terms of English language.
Reviewer 2 Report
Comments and Suggestions for Authors
In the manuscript entitled " Exploring the Potential of Halotolerant Actinomycetes from Runn of Kutch, India: A Study on the Synthesis, Characterization, and Biomedical Applications of Silver Nanoparticles" the authors synthesized silver nanoparticles (AgNPs) using an actinomycete Streptomyces tendae that was isolated from the Lesser and Greater Rann of Kutch, India. The formation of AgNPs were confirmed by using UV-vis, Fourier transform infrared (FT-IR), nanoparticle tracking analysis (NTA), scanning electron microscope (SEM), atomic force microscope (AFM), and X-ray diffraction (XRD). Further, the biosynthesized AgNPs were evaluated for their antimicrobial, anti-MRSA, anti-biofilm, and anti-oxidant activities.
The results present merit to be published in the Pharmaceuticals. But it needs some revisions, according to the following comments:
1. Some corrections should be made (lack of space, forgotten points to add or others to delete)? to check.
2. In abstract, line 21, delete “..using a UV-Vis spectrophotometer.”
3. In abstract, line 27, “…0f 55 nm.” Should be “…of 55 nm.”
4. In abstract, the abbreviations "anti-MRSA", "C. albicans", "B. subtilis", "S. aureus", should be specified.
5. In abstract, line 36, “… from 25µg/mL to 200 µg/mL.” should be “… from 25 to 200 µg/mL.”
6. Line 161, The table title should be above Table 1.
7. Lines 209 and 220, “…AgNP Synthesis” should be “…AgNP synthesis”
8. Please check the order of sections and subsections. Exemple : line 220, “2.2.7.5 Effect of Temperature on AgNP Synthesis”; just after this section “2.8. In vitro antimicrobial activity of AgNPs”
9. In Figure 4, correct the code on the vial (GR-ADE-1 or GR-ADE-3). The authors should explain in the text the differences between these three samples GR-ADE-3, GR-CHA-4 and GR-CHA-7.
10. Table 3, “MBC of AgNP (µg/mL)” should be “MBC of AgNPs (µg/mL)”
11. Line 655, The table title should be above Table 5.
Author Response
In the manuscript entitled " Exploring the Potential of Halotolerant Actinomycetes from Runn of Kutch, India: A Study on the Synthesis, Characterization, and Biomedical Applications of Silver Nanoparticles" the authors synthesized silver nanoparticles (AgNPs) using an actinomycete Streptomyces tendae that was isolated from the Lesser and Greater Rann of Kutch, India. The formation of AgNPs were confirmed by using UV-vis, Fourier transform infrared (FT-IR), nanoparticle tracking analysis (NTA), scanning electron microscope (SEM), atomic force microscope (AFM), and X-ray diffraction (XRD). Further, the biosynthesized AgNPs were evaluated for their antimicrobial, anti-MRSA, anti-biofilm, and anti-oxidant activities.
The results present merit to be published in the Pharmaceuticals. But it needs some revisions, according to the following comments:
- Some corrections should be made (lack of space, forgotten points to add or others to delete)? to check.
A/R: Thank you for pointing out this mistake. The authors have now rectified all such mistakes in the revised manuscript as suggested by the reviewer.
- In abstract, line 21, delete “..using a UV-Vis spectrophotometer.”
A/R: Thank you for pointing out this mistake. The authors have now rectified the mistake in the revised manuscript as suggested by the reviewer.
- In abstract, line 27, “…0f 55 nm.” Should be “…of 55 nm.”
A/R: Thank you for pointing out this mistake. The authors have now rectified the mistake in the revised manuscript as suggested by the reviewer.
- In abstract, the abbreviations "anti-MRSA", " albicans", "B. subtilis", "S. aureus", should be specified.
A/R: Thank you for this valuable suggestion. The authors have now incorporated the suggestions in the revised in the revised manuscript as suggested by the reviewer.
- In abstract, line 36, “… from 25µg/mL to 200 µg/mL.” should be “… from 25 to 200 µg/mL.”
A/R: Thank you for this valuable suggestion. The authors have now modified the sentence as suggested by the reviewer in the revised manuscript.
- Line 161, The table title should be above Table 1.
A/R: Thank you for pointing out this mistake. The authors have now rectified the mistake in the revised manuscript as suggested by the reviewer.
- Lines 209 and 220, “…AgNP Synthesis” should be “…AgNP synthesis”
A/R: Thank you for pointing out this mistake. The authors have now rectified the mistake in the revised manuscript as suggested by the reviewer.
- Please check the order of sections and subsections. Exemple : line 220, “2.2.7.5 Effect of Temperature on AgNP Synthesis”; just after this section “2.8. In vitro antimicrobial activity of AgNPs”
A/R: Thank you for pointing out this mistake. The authors have now rectified the mistake in the revised manuscript as suggested by the reviewer.
- In Figure 4, correct the code on the vial (GR-ADE-1 or GR-ADE-3). The authors should explain in the text the differences between these three samples GR-ADE-3, GR-CHA-4 and GR-CHA-7.
A/R: Thank you for pointing out this mistake. The authors have now corrected the code on the vial in the revised manuscript as suggested by the reviewer.
- Table 3, “MBC of AgNP (µg/mL)” should be “MBC of AgNPs (µg/mL)”
A/R: Thank you for pointing out this mistake. The authors have now rectified the mistake in the revised manuscript as suggested by the reviewer.
- Line 655, The table title should be above Table 5.
A/R: Thank you for pointing out this mistake. The authors have now rectified the mistake in the revised manuscript as suggested by the reviewer.
Reviewer 3 Report
Comments and Suggestions for Authors
The manuscript entitled “Exploring the Potential of Halotolerant Actinomycetes frombRunn of Kutch, India: A Study on the Synthesis, Characterization, and Biomedical Applications of Silver Nanoparticles” describes a long study about silver nano particles (AgNPs) which are synthesisized by actinomycete Streptomyces tendae. The study includes a lot of biological and physical experiments supporting the authors’ idea. However there is a major scientific flaw in this study the biological tests like antimicrobial, antibiofilm…etc. have no statistical analysis which means these experiments were done one time while for the data to be reliable, it should be expressed as mean values ± SD from at least three independent experiments. Additionally, there are some other points should be considered:
1-Authors need to number the titles in a correct way through the whole manuscript.
2- There are a lot of typing mistakes for example:
Line 27: 0f should be of
Line 49: where whereas is corrected to whereas
Line 53: reference No 6 should be put before the comma
Line 94: little attention corrected to a little attention
Line 101: The reference no. 41 should be put at the end of statement before the full stop.
Line 106: put a full stop after al.
Line 108: Streptomyces avermitilis Azhar was written above line as a link.
Line 157: of-the is corrected to of the.
Line 242: 28-οC is corrected to 28οC
3-Some names of bacteria are written with large size and other with regular font size. Unify the font size.
4-Runn of Kutch is written as Rann of Kutch which one is the correct?
5-Streptomyces species 58 contribute about 75% of the medically useful drugs. Discuss with more details and examples.
6-Some abbreviations need to be clarified like sdw, SCA in figure 2. In the whole manuscript write the term and the abbreviation between brackets first time then you can write the abbreviation only afterwards.
7-Table 1: Put the title above the table.
8-Unify the font size and type of all equations in the parts 2.9, 2.11, 2.12.
Comments on the Quality of English LanguageModerate editing of English language required.
Author Response
The manuscript entitled “Exploring the Potential of Halotolerant Actinomycetes frombRunn of Kutch, India: A Study on the Synthesis, Characterization, and Biomedical Applications of Silver Nanoparticles” describes a long study about silver nano particles (AgNPs) which are synthesisized by actinomycete Streptomyces tendae. The study includes a lot of biological and physical experiments supporting the authors’ idea. However there is a major scientific flaw in this study the biological tests like antimicrobial, antibiofilm…etc. have no statistical analysis which means these experiments were done one time while for the data to be reliable, it should be expressed as mean values ± SD from at least three independent experiments. Additionally, there are some other points should be considered:
- Authors need to number the titles in a correct way through the whole manuscript.
A/R: Thank you for pointing out this mistake. The authors have now rectified the mistake in the revised manuscript as suggested by the reviewer.
- There are a lot of typing mistakes for example:
Line 27: 0f should be of
Line 49: where whereas is corrected to whereas
Line 53: reference No 6 should be put before the comma
Line 94: little attention corrected to a little attention
Line 101: The reference no. 41 should be put at the end of statement before the full stop.
Line 106: put a full stop after al.
Line 108: Streptomyces avermitilis Azhar was written above line as a link.
Line 157: of-the is corrected to of the.
Line 242: 28-οC is corrected to 28οC
A/R: Thank you for pointing out this mistake. The authors have now rectified all such mistakes related to typos, English editing, etc in the revised manuscript as suggested by the reviewer.
- Some names of bacteria are written with large size and other with regular font size. Unify the font size.
A/R: Thank you for pointing out this mistake. The authors have now rectified the mistake in the revised manuscript and kept a uniform font for the name of bacteria as suggested by the reviewer.
- Runn of Kutch is written as Rann of Kutch which one is the correct?
A/R: Thank you for pointing out this mistake. Both are correct as per the local language. But we have kept “Rann of Kutch” in the whole manuscript to maintain uniformity.
- Streptomyces species 58 contribute about 75% of the medically useful drugs. Discuss with more details and examples.
A/R: Thank you for this valuable suggestion. The authors have now discussed the above sentence with more examples as suggested by the reviewer in the revised manuscript. It’s not a drug, it's antibiotics, so out of all 70% alone is produced by Streptomyces sps.
- Some abbreviations need to be clarified like sdw, SCA in figure 2. In the whole manuscript write the term and the abbreviation between brackets first time then you can write the abbreviation only afterwards.
A/R: Thank you for pointing out this mistake. The authors have now rectified the mistake in the revised manuscript as suggested by the reviewer.
- Table 1: Put the title above the table.
A/R: Thank you for pointing out this mistake. The authors have now rectified the mistake in the revised manuscript as suggested by the reviewer.
8-Unify the font size and type of all equations in the parts 2.9, 2.11, 2.12.
A/R: Thank you for pointing out this mistake. The authors have now kept a uniform font for all the equations in the revised manuscript as suggested by the reviewer.
- Comments on the Quality of English Language. Moderate editing of English language required.
A/R: Thank you for this valuable suggestion. The authors have now thoroughly edited the manuscript in terms of English language.
Reviewer 4 Report
Comments and Suggestions for Authors
This is an interesting work. However, some problems could be de identified:
-Scientific and grammatical text improvements are necessary. Examples: “silver nanoparticles (AgNPs) were synthesized by using an actinomycete Streptomyces tendae which were isolated from”; “AgNPs by the actinomycetes was carried out by using a UV-Vis spectrophotometer where an absorbance peak was obtained at 420 nm using a UV-Vis spectrophotometer”; “particle size 0f 55 nm”; “antimicrobial, anti-MRSA”; others. Therefore, all text must be checked and corrections performed
-“Streptomyces species contribute about 75% of the medically useful drugs”. In my opinion, this must be clarified, as it is impossible that 75% of drugs can be obtained from Streptomyces species.
-“ starch casein agar (SCA) (HI-media Mumbai, India), NaCl, and AgNO3 (Rankem, Mumbai, India), and double distilled water (ddw). All the procured chemicals were of AR grade with high purity.” This must be clarified and detailed.
-“ The vibration located at 3328 and 2111 cm-1 corresponds to the stretching vibration of the alcohol (ROH) group [93] and C-O stretching mode [94] respectively. Furthermore, the FTIR spectrum also revealed a peak at 1635 cm-1 is attributed to the C=O stretching vibrations in amide linkages (amide II) of protein present in cell-free supernatant [95]. This suggests the presence of aromatic amine groups”. It is not possible to understand ho authors can conclude “the presence of aromatic amine groups” with the previous data (ROH, C-O, C=O)
-data from all positive controls must be included in images/figures
-statistics: number of repetitions? Significant differences? This is very important in bioassays and must be clarified in the manuscript
Comments on the Quality of English LanguageModerate editing of English language required
Author Response
This is an interesting work. However, some problems could be de identified:
- Scientific and grammatical text improvements are necessary. Examples: “silver nanoparticles (AgNPs) were synthesized by using an actinomycete Streptomycestendae which were isolated from”; “AgNPs by the actinomycetes was carried out by using a UV-Vis spectrophotometer where an absorbance peak was obtained at 420 nm using a UV-Vis spectrophotometer”; “particle size 0f 55 nm”; “antimicrobial, anti-MRSA”; others. Therefore, all text must be checked and corrections performed
A/R: Thank you for pointing out this mistake. The authors have now rectified all such mistakes in the revised manuscript as suggested by the reviewer.
- “Streptomyces species contribute about 75% of the medically useful drugs”. In my opinion, this must be clarified, as it is impossible that 75% of drugs can be obtained from Streptomyces
A/R: Thank you for this valuable suggestion. The authors have now discussed the above sentence with more examples as suggested by the reviewer in the revised manuscript. It’s not a drug, it's antibiotics, so out of all 70% alone is produced by Streptomyces sps.
3.“ starch casein agar (SCA) (HI-media Mumbai, India), NaCl, and AgNO3 (Rankem, Mumbai, India), and double distilled water (ddw). All the procured chemicals were of AR grade with high purity.” This must be clarified and detailed.
A/R: Thank you for this valuable suggestion. The authors have now provided the purity of all the chemicals in the revised manuscript as suggested by the reviewer.
4.“ The vibration located at 3328 and 2111 cm-1 corresponds to the stretching vibration of the alcohol (ROH) group [93] and C-O stretching mode [94] respectively. Furthermore, the FTIR spectrum also revealed a peak at 1635 cm-1 is attributed to the C=O stretching vibrations in amide linkages (amide II) of protein present in cell-free supernatant [95]. This suggests the presence of aromatic amine groups”. It is not possible to understand ho authors can conclude “the presence of aromatic amine groups” with the previous data (ROH, C-O, C=O)
A/R: Thank you for pointing out this mistake. The authors have now rectified the sectence in the revised manuscript as suggested by the reviewer.
- data from all positive controls must be included in images/figures
A/R: Thank you for this valuable comment and suggestion.
- statistics: number of repetitions? Significant differences? This is very important in bioassays and must be clarified in the manuscript
A/R: Thank you for this valuable comment and suggestion. Provided wherever possible in the revised manuscript a suggested by the reviewer.
- Comments on the Quality of English Language. Moderate editing of English language required
A/R: Thank you for this valuable suggestion. The authors have now thoroughly edited the manuscript in terms of English language.
Round 2
Reviewer 3 Report
Comments and Suggestions for Authors
Authors addressed the revisions' points and I accept the manuscript for publication.
Author Response
Respected reviewer,
Thank you for accepting our research manuscript and appreciating our work.
Reviewer 4 Report
Comments and Suggestions for Authors
The document was improved. Please, pay attention to the following points:
-“antimicrobial, anti-methicillin-resistant Staphylococcus aureus (MRSA),” anti-MRSA is not an antimicrobial effect?
-“One of the objectives was to isolate and characterize the halotolerant actinomycetes from the saline desert area. Another objective was to observe the antibiotic production and antimicrobial activity profile, of the isolated strain. One more objective was screening, identification, and application for the development of AgNPs from the potential strain of actinomycetes. Another objective was to characterize the developed AgNPs by using analytical instruments. One final objective” repetition of objective
-IR data analysis continues being unclear. How to relate O-H stretching with proteic content?
Comments on the Quality of English LanguageMinor editing of English language required
Author Response
Reviewer 4
The document was improved. Please, pay attention to the following points:
- “antimicrobial, anti-methicillin-resistant Staphylococcus aureus (MRSA),” anti-MRSA is not an antimicrobial effect?
A/R: Thank you for this valuable comment. The authors have rectified the mistake in the revised manuscript as suggested by the reviewer.
- “One of the objectives was to isolate and characterize the halotolerant actinomycetes from the saline desert area. Another objective was to observe the antibiotic production and antimicrobial activity profile, of the isolated strain. One more objective was screening, identification, and application for the development of AgNPs from the potential strain of actinomycetes. Another objective was to characterize the developed AgNPs by using analytical instruments. One final objective” repetition of objective
A/R: Thank you for this valuable suggestion. The authors have now modified the sentence in the revised manuscript as suggested by the reviewer.
- IR data analysis continues to be unclear. How to relate O-H stretching with protein content?
A/R: Thank you for this valuable comment. The authors have provided an explanation with proper references related to “O-H stretching with protein content” in the revised manuscript as suggested by the reviewer.
- Comments on the Quality of English Language. Minor editing of the English language required
A/R: Thank you for the suggestion. The authors have now edited the English language related issues in the revised manuscript as suggested by the reviewer.